# The Role and Prospects of Mesenchymal Stem Cells in Skin Repair and Regeneration

**DOI:** 10.3390/biomedicines12040743

**Published:** 2024-03-27

**Authors:** Si Wu, Shengbo Sun, Wentao Fu, Zhengyang Yang, Hongwei Yao, Zhongtao Zhang

**Affiliations:** 1Department of General Surgery, Beijing Friendship Hospital, Capital Medical University, Beijing 100050, China; 2National Clinical Research Center for Digestive Diseases, Beijing 100050, China; 3School of Basic Medical Sciences, Capital Medical University, Beijing 100050, China

**Keywords:** mesenchymal stem cells, wound healing, skin regeneration, skin rejuvenation

## Abstract

Mesenchymal stem cells (MSCs) have been recognized as a cell therapy with the potential to promote skin healing. MSCs, with their multipotent differentiation ability, can generate various cells related to wound healing, such as dermal fibroblasts (DFs), endothelial cells, and keratinocytes. In addition, MSCs promote neovascularization, cellular regeneration, and tissue healing through mechanisms including paracrine and autocrine signaling. Due to these characteristics, MSCs have been extensively studied in the context of burn healing and chronic wound repair. Furthermore, during the investigation of MSCs, their unique roles in skin aging and scarless healing have also been discovered. In this review, we summarize the mechanisms by which MSCs promote wound healing and discuss the recent findings from preclinical and clinical studies. We also explore strategies to enhance the therapeutic effects of MSCs. Moreover, we discuss the emerging trend of combining MSCs with tissue engineering techniques, leveraging the advantages of MSCs and tissue engineering materials, such as biodegradable scaffolds and hydrogels, to enhance the skin repair capacity of MSCs. Additionally, we highlight the potential of using paracrine and autocrine characteristics of MSCs to explore cell-free therapies as a future direction in stem cell-based treatments, further demonstrating the clinical and regenerative aesthetic applications of MSCs in skin repair and regeneration.

## 1. Introduction

MSCs are a multipotent type of stem cells that originate from the mesoderm and ectoderm [1] in early development and can differentiate into a wide range of tissue cells, such as bone, cartilage, fat, muscle, and nerve cells. MSCs were first discovered in bone marrow, but have since been found in many other tissues of the body, such as adipose tissue, synovial tissue bone, muscle, lung, liver, pancreas, amniotic fluid, and umbilical cord blood [2,3]. MSCs from different sources differ in their accessibility, content, proliferation ability, immunomodulatory capacity, and the cytokines they secrete, and have different therapeutic potentials for different diseases.

Since MSCs were first discovered in bone marrow by Cohnheim in 1867 [4], they have been considered to play an important role in skin regeneration and wound healing. In 1991, Professor Caplan first introduced the concept of mesenchymal stem cells and emphasized their potential for multidirectional differentiation [5]. Subsequently, many researchers have isolated MSCs from different tissues and have demonstrated that they can differentiate into a wide range of cell types such as osteoblasts, adipocytes, chondrocytes, tenocytes, cardiomyocytes, keratinocytes, hepatocytes, and neural cells [6,7]. For example, dental-derived MSCs can be obtained from different parts of the teeth, such as the pulp, the ligament, the follicle, and the gingiva [8]. These stem cells have the ability to differentiate into various tissue types, such as bone, cartilage, fat, nerve, and skin [8]. Dental-derived MSCs can be used to repair damaged tissues and organs, and to treat diseases that; affect the immune system, the nervous system, the liver, and the skin [9]. In addition, MSCs have been found to have various biological functions such as immunomodulation, anti-inflammation, anti-apoptosis, and pro-angiogenesis [10,11]. In 1995, Professor Caplan extracted and isolated cultured MSCs from the bone marrow of patients with malignant hematological disorders and then infused them back into the patients to observe the clinical effects and to demonstrate the safety of these matrices [12]. Since then, the clinical application of MSCs has gradually expanded to a wide range of diseases, such as cardiovascular diseases, neurological diseases, bone and joint diseases, autoimmune diseases, liver diseases, diabetes, and so on [13].

Currently, several MSC-based therapies have received FDA approval for the treatment of acute graft-versus-host disease, bone defects, osteoporotic vertebral compression fractures, and ischemic heart failure [14]. According to ClinicalTrials.gov clinical registry data, there are currently more than 1300 clinical trials related to MSCs around the world, covering more than 300 diseases. Most of these clinical trials are being carried out for musculoskeletal disorders, central nervous system disorders, immune system disorders, wounds and traumas, rheumatic disorders, joint disorders, arthritis, vascular disorders, respiratory disorders, digestive disorders, and gastrointestinal disorders. These clinical studies have demonstrated that MSCs are a safe and effective therapeutic tool, providing a new treatment strategy for difficult multi-system diseases [15].

The clinical value of MSCs is not only reflected in their therapeutic efficacy, but also in their advantages in terms of high proliferation ability, immunogenicity, and differentiation capacity [16]. Currently, the most used stem cells are bone marrow-derived mesenchymal stem cells, but they have some limitations, such as a decline in the number and activity of the stem cells with age [17]. Therefore, the search for other alternative sources of MSCs is an important research direction. Among them, umbilical cord-derived MSCs have some advantages, such as abundant number of stem cells, easy collection, no harm to donors, low immunogenicity, high differentiation capacity, and no ethical controversy [18]. Umbilical cord-derived MSCs have been used to treat a wide range of diseases, such as cardiovascular diseases, liver diseases, bone and muscle degenerative diseases, neurological injuries, and autoimmune diseases [19,20,21].

In conclusion, MSCs are stem cells with a wide range of clinical applications, and they can promote tissue and organ regeneration and repair through a variety of mechanisms, providing new possibilities for the treatment of a wide range of diseases [13]. However, the clinical application of MSCs also faces some challenges and problems, such as cell quality control, immunological rejection, carcinogenic risk, standardized management, etc., which require further research and optimization [22].

## 2. The Wound Healing-Promoting Mechanisms of Mesenchymal Stem Cells

Wound healing is a complex biological process involving the coordinated action of multiple cellular, molecular, and signaling pathways. The process of wound healing can be divided into four phases: hemostasis, inflammation, proliferation, and remodeling (Figure 1) [23,24]. Under normal circumstances, the wound healing process is orderly and can restore the structure and function of the tissue. However, in some cases of disease or injury, the process of wound healing may be disrupted, resulting in difficulty in healing or the formation of pathological scars [25]. Therefore, exploring effective ways to promote wound healing and improve scar quality is an important clinical issue.

Stem cells are a class of cells with self-renewal and multidirectional differentiation capabilities, and they play an important role in tissue repair and regeneration [26]. MSCs are adult stem cells found in a wide range of tissues; Figure 2 provides an illustration of the sources of mesenchymal stem cells [27]. The low immunogenicity, ease of isolation and expansion, multiple differentiation potentials, and paracrine functions of MSCs make them ideal candidates for trauma therapy [28]. In recent years, more and more studies have shown that MSCs can promote wound healing through various mechanisms, such as through their differentiation potential, paracrine function, and promotion of angiogenesis [29,30]. Among them, the MSCs’ paracrine function refers to their involvement in the inflammatory, proliferative, and remodeling processes of wound healing by secreting a variety of bioactive molecules, such as growth factors, cytokines, chemokines, exosomes, etc., to influence the function and state of the surrounding cells and tissues [31]. Neovascularization is the division of new blood vessels from the surrounding normal blood vessels during the wound healing process due to the inflammatory response and angiogenic factors [32]. The proliferation and migration of endothelial cells in a wound form a new vascular network, thereby improving blood supply and oxygen delivery to the wound and promoting wound healing and tissue regeneration [33]. The newborn blood vessels can provide oxygen and nutrients to the wound tissue, promote cell proliferation and differentiation, and enhance the resistance and repair ability of the wound [32]. The newborn blood vessels also release some cytokines, such as vascular endothelial growth factor (VEGF), transforming growth factor-β (TGF-β), IL-6, etc., which regulate the inflammatory response and fibrotic process, inhibit the activation and secretion of inflammatory cells, and promote the proliferation of fibroblasts and collagen synthesis [31,34]. In addition, neovascularization improves the microcirculation in the wound tissue, reduces tissue hypoxia and edema, reduces the damage to and death of capillary endothelial cells, and maintains capillary patency [35]. The newborn blood vessels can influence the remodeling process of the wound tissue, promoting steps such as epithelial formation, collagen remodeling, fibrosis, and the formation of structures such as granulation tissue and scar tissue [36,37]. In conclusion, neovascularization plays an important role in wound healing as both a stimulating and regulatory factor. Neovascularization interacts with other factors that, together, determine the efficiency and quality of wound healing.

MSCs can induce the proliferation, migration, and differentiation of peripheral vascular endothelial cells (VECs) by directly contacting or indirectly acting on VECs, thus promoting the formation and maturation of neovascularization of wounds, and improving the wound healing rate and functional recovery [38]. Endothelial cells are the main constituent cells of blood vessels and they play a key role in the process of neovascularization in wounds. Direct contact refers to the presence of physical or chemical interactions between MSCs and VECs, such as mechanical stretching, shear stress, and pharmacological inhibitors, which can enhance signaling and silencing of transcription factors between VECs and MSCs [39]. Indirect effects are defined as the effects of MSCs on the microenvironment around VECs through the release of exosomes or other factors, such as vascular endothelial growth factor A (VEGFA), epidermal growth factor (EGF), and fibroblast growth factor 2 (FGF2) [40,41]. The exosomes secreted by MSCs can bind to receptors on VECs, activating downstream signaling pathways such as the EGFR, ERK1/2, PI3K/Akt pathways, etc., and regulating the proliferation, migration, and differentiation of VECs, thereby promoting neovascularization [41].

MSCs are a class of adult stem cells with multiple differentiation potentials, which can differentiate into skin tissue cells, such as keratinocytes, fibroblasts, endothelial cells, etc., under the appropriate conditions, and thus participate in the re-epithelialization of wounds, formation of granulation tissue, and regeneration of blood vessels [37]. Keratinocytes are the main cell type of the skin’s surface layer and they play a key role in the re-epithelialization of wounds [42]. Re-epithelialization is the migration and proliferation of epithelial cells from the surface of the wound towards the center of the wound to form a new epithelial layer and restore the barrier function of the skin [43]. It has been found that exosomes isolated from human umbilical cord mesenchymal stem cells can accelerate re-epithelialization of burn wounds by enhancing the downstream effects of Wnt signaling through the increased nuclear translocation of β-catenin, thereby promoting the proliferation and migration of keratin-forming cells [44,45]. Exosomes are vesicles encapsulated by cell membranes, which can carry a wide range of biologically active molecules, such as proteins, nucleic acids, and lipids, thereby transmitting information between cells [46]. The Wnt/β-catenin signaling pathways are signal transduction pathways involved in cell proliferation, and play an important role in skin development and regeneration [47]. In recent years, it has been demonstrated that human umbilical cord mesenchymal stem cell exosomes could also accelerate the re-epithelialization of burn wounds by increasing the phosphorylation level of AKT and enhancing the downstream effects of AKT signaling, thereby promoting the proliferation and migration of keratin-forming cells [46]. The AKT signaling pathway is a signal transduction pathway involved in cell survival and proliferation, and plays an important role in skin regeneration [38]. Fibroblasts are the main cell type in the dermis of the skin and they play an important role in the proliferation and remodeling stages of wound healing [48]. Proliferation and remodeling refer to the synthesis and secretion of large amounts of extracellular matrix, such as collagen, elastic fibers, fibronectin, etc., by fibroblasts in the wound, thereby forming granulation tissue that fills in the wound defect and strengthens the wound [48]. It has been demonstrated that exosomes isolated from human adipose MSCs can reduce the proliferation of scar fibroblasts and collagen synthesis by inhibiting the TGF-β/Smad signaling pathway and improve scar formation in burn wounds [49]. Keloid fibroblasts are a specific type of fibroblasts that appear during the wound healing process; they are highly proliferative and secretory but lack the ability to degrade and remodel the extracellular matrix, leading to the excessive deposition of extracellular matrix and fibrosis of the tissue [50]. The TGF-β/Smad signaling pathway is a signal transduction pathway involved in cell proliferation and differentiation, and plays an important role in the regulation of scar formation [51]. Studies have shown that human adipose MSC exosomes can reduce scar formation in burn wounds by decreasing the expression and activity of TGF-β and inhibiting the phosphorylation and nuclear translocation of Smad2/3, thereby reducing scar fibroblast proliferation and collagen synthesis [52]. Additionally, MSCs also crosstalk with tissue cells such as macrophages to promote wound healing. Macrophages play a beneficial role in the wound repair process, and the anti-inflammatory M2 phenotype promotes wound healing in the later stages of wound healing [53]. Exosomes of MSCs are able to promote macrophage M2 polarization by targeting pknox1 so as to enhance wound healing [54].

## 3. The Potential of Mesenchymal Stem Cells in the Treatment of Clinical Diseases

The use of MSCs in clinical diseases is a cutting-edge research area that promises to provide new strategies and approaches for the treatment of a variety of skin injuries and diseases, such as chronic refractory wounds and burn injuries. The main aspects of acute and chronic wound healing are at the anatomical level.

MSC-derived extracellular vesicles (EVs) aid in tissue regeneration by facilitating the regrowth of the epidermis, dermis, hair follicles, nerves, and blood vessels, while also mitigating abnormal pigmentation (Figure 3) [55]. The roles of MSCs in wound healing are shown in Table 1. Chronic refractory wounds mean that the wounds cannot achieve structural and functional integrity in time and eventually cause a chronic inflammatory state [56]. They usually require long-term care and treatment to promote healing and prevent complications. The difficulty in the treatment of chronic refractory wounds lies in the lack of effective stimulating and growth factors, as well as the lack of precise control and regulation of the wound tissue. Burn injuries involve damage to the skin tissues due to friction, cold, heat, radiation, chemical, or electric sources [57]. They usually cause severe complications such as edema, necrosis, and infections [57]. The difficulties in the treatment of burn injuries lie in the lack of effective protective barriers and repair mechanisms, as well as the lack of individualized therapeutic treatments for different types and degrees of burns. In this section, we focus on the potential of MSCs in the treatment of clinical diseases from two perspectives: the treatment of chronic refractory wounds and burn injuries.

To date, the treatments for chronic refractory wounds have included debridement, topical antibiotics, compression bandages, skin grafts, and cytokines [67,68,69,70]. However, all of these methods have certain limitations and side effects, for example, debridement can cause infection and bleeding [71], etc. Therefore, there is an urgent need to explore some new approaches. MSC therapy is a novel therapy with great potential to restart the normal healing response in old wounds by increasing the accumulation of MSCs in the wound. The mechanisms of stem cell effects in chronic refractory wound healing include cell recruitment, cell differentiation, immunomodulation, antimicrobial effects, pro-angiogenic effects, and epidermal replantation [72,73]. Stem cells are used in chronic refractory wounds by directly injecting MSCs into the wound [74,75]. Direct injection can rapidly increase the number and distribution of MSCs, promoting wound healing and tissue regeneration [75]. Several clinical trials and studies have been conducted to investigate the efficacy and safety of MSCs in the treatment of chronic refractory wounds [55]. In short, MSCs have great potential and advantages in the treatment of chronic refractory wounds, and can be used in a variety of ways to achieve repair of and tissue regeneration in wounds. However, there are still some challenges and problems, such as the need for the further optimization and validation of stem cell sources, quality, quantity, preservation, transport, and safety. More high-quality clinical trials are needed in the future to assess the efficacy and safety of MSCs in different types and degrees of chronic refractory wounds and to explore the optimal use and dosage.

Besides their superiority in treatments for chronic refractory wounds, MSCs also have great potential in the treatment of burn injuries and can promote the healing of burn wounds in several ways. Firstly, MSCs can inhibit the excessive inflammatory response after a burn injury by reducing the infiltration of inflammatory cells and releasing inflammatory factors, thus reducing the severity of the burn injury and the risk of complications [76]. They can also trigger the polarization of macrophages from the pro-inflammatory M1 type to pro-healing M2 type, thereby promoting wound cleaning and repair [77]. In addition, MSCs can stimulate the formation of neovascularization by secreting growth factors such as VEGF and FGF, which increase blood perfusion and provide nutrients to burn wounds, thereby accelerating wound healing [78]. They can improve the structure and function of wounds by secreting matrix proteins such as collagen, elastin, and fibronectin, which promote the reconstruction of the dermis and enhance the strength and elasticity of the wound [79]. MSCs can also stimulate the proliferation and differentiation of epidermal cells and promote the regeneration of the epidermal layer through the secretion of growth factors such as transforming growth factor-β (TGF-β), epidermal growth factor (EGF), and keratinocyte growth factor (KGF), thereby restoring the barrier function of the skin [23]. Furthermore, MSCs can regulate the remodeling of the extracellular matrix (ECM) by secreting enzymes such as matrix metalloproteinases (MMPs) and tissue inhibitors of metalloproteinases (TIMPs), which balances matrix synthesis and degradation, thereby reducing scar formation after a burn [80]. They can also inhibit keloid formation by restraining the proliferation and secretion of fibroblasts, thus improving the restoration effects [81].

In summary, MSCs have multifaceted advantages in the treatment of burns and can influence the healing process of burn wounds at multiple levels, including inflammation, vascularity, and scarring, in the dermis and epidermis to improve the quality of life and survival rate of burn patients. Currently, several clinical studies and case reports have confirmed the efficacy and safety of MSCs in the treatment of burn injuries [82], but further exploration of the optimal source, dosage, delivery, timing, and mechanism of MSCs is still needed to provide a more optimal treatment plan for burn patients.

## 4. Applications of Mesenchymal Stem Cells in Skin Regeneration and Rejuvenation

The use of the regeneration ability of MSCs is not limited to disease treatments and is also used in the aesthetic field. MSCs have received attention due to their multilineage differentiation abilities. Some studies have reported that mesenchymal stem cell exosomes can be utilized in the treatment of skin aging. Except for chronological aging, human skin also undergoes photoaging, a kind of sun-induced skin aging. Actually, in the past 30 years, the research on the molecular mechanisms of skin photoaging such as the production of intracellular ROS has made substantial progress. The generation of reactive oxygen species (ROS) damages the connective tissues in human skin [83]. The exposure to ultraviolet rays for a long time has been proven to decrease collagen protein production.

In contrast, MSCs, with their antioxidation and anti-apoptosis effects, can reduce the production of metalloproteinases (MMPs) and activate the proliferation of dermal fibroblasts [84]. A study compared adipose-derived stem cells (ADSCs) with fibroblasts in the improvement of skin wrinkles caused by photoaging and found that ADSCs are as effective as fibroblasts in promoting the production of collagen protein [85]. There are plenty of studies that have demonstrated that therapies and medicines based on stem cells can inhibit the signaling cascades which participate in telomere shortening, estrogen depletion, and excessive ROS production [86]. It has been reported that ADSCs have anti-aging effects on aging cells and in animal models of premature aging. ADSCs can accelerate mitophagy, eliminate intracellular ROS, and eventually improve the number of mitochondria [87]. Exosomes from the conditioned medium of human induced pluripotent stem cells (iPSCs) have a therapeutic potential in the treatment of skin aging caused by photoaging and natural senescence. Pretreated induced pluripotent stem cells exosomes (iPSCs-Exo’s) can inhibit the overexpression of MMP-1/3 and attenuate the human dermal fibroblast (HDF) injury caused by UVB and finally restore the expression of type I collagen [88]. By injecting autologous adipose-derived mesenchymal stem cells expanded in vitro, completely de novo oxytalan and elaunin fiber production in the subepidermal area and the rebuilding of the dermis–epidermal junction structure were observed, indicating the complete rescue of solar elastic deformation [89].

Moreover, adipose stem cell-conditioned culture medium (ADSC-CM) can reduce UVB-induced apoptosis and stimulate collagen synthesis by HDFs to reduce the appearance of wrinkles, which was confirmed by a shortening of the subG1 phase of HDFs. ADSC-CM can also increase the expression of type I collagen in HDFs and increase the level of metalloproteinase 1 [90]. Conditioned media from human umbilical cord blood-derived mesenchymal stem cells (USC-CM) contains epithelial growth factor (EGF), basic fibroblast growth factor (bFGF), platelet-derived growth factor (PDGF), hepatocyte growth factor (HGF), collagen type 1, and growth differentiation factor-11 (GDF-11), one of the youth growth factors [91]. In an experiment, GDF-11 significantly accelerated the growth and migration of HDFs and also improved the production of extracellular matrix (ECM). USC-CM exosomes (USC-CM Exo’s) contain the growth factor related to skin rejuvenation which acts on HDFs to promote cell migration and collagen synthesis [92].

In recent years, the extracts secreted by mesenchymal stem cells have been used as a prospective biological therapy. However, the extracts are unstable and non-specific which could be resolved using artificial intelligence. Due to the ability of machine learning and artificial intelligence to predict and simulate protein folding and peptide/protein structure interactions, they have been used for screening promising biomimetic peptide components in mesenchymal stem cell secretions. One peptide virtual screening model which identified EQ-9 as a peptide with anti-aging and skin repair functions. EQ-9 potentially inhibits inflammation through increasing the fibroblast survival rate and decreasing intracellular ROS levels [93]. It has been reported that microfragmented adipose tissue containing adipose stem cells combined with a crosslinked hyaluronic acid scaffold can improve soft tissue defects such as deep wrinkles [94].

## 5. Methods for Enhancing the Effects of MSCs

### 5.1. Stem Cell Delivery into Skin Wounds

A dermal substitute is able to promote wound healing by shortening the healing time and improving the impaired function of the injured tissues. As a prospective carrier of stem cells, SECM-MC hydrogel, composed of soluble ECM (sECM) and methyl cellulose (MC), was inserted into a full-layer skin wound, which led to wound healing through re-epithelialization and neovascularization [95].

Through the combination of biomaterials and living cells, tissue engineering technology can promote the development of regenerative medicine. Tissue engineering scaffolds can transport stem cells to the sites in need of repair, eventually increasing the retention and implantation rates of stem cell transplants [96,97]. The rise of 3D printing technology further enriches the structural design and composition of tissue engineering scaffolds, and also provides convenience for the loading and delivery of live cells. Transplanting human gingival tissue pluripotent mesenchymal stem cells/stromal cells in 3Dprinted medical grade polycaprolactone (mPCL) dressings resulted in wound contracture and significantly improved skin regeneration through granulation and re-epithelialization [98]. One study showed that dermal vascular endothelial cells in wounds treated with chitosan exhibited better immunoreactivity [99]. A new type of chitosan/decellularized dermal matrix (CHS/ADM) stem cell delivery system can overcome the limitation of the traditional collagen delivery system, which lacks a response to high ROS environments. In a high ROS microenvironment, the new system acts as a protective screen and effectively clears a certain amount of the ROS, protecting mesenchymal stem cells (MSCs) from oxidative stress [100]. Mesenchymal stem cells derived from rat adipose tissue seeded on collagen–chitosan scaffolds and implanted into wounds can completely heal the wound, restoring the epidermis and dermis to a normal state [101]. Decellularized human amniotic membranes (dAMs) and matrix (sAM) were used as wound dressing scaffolds. AdMSCs were seeded onto dAMs or sAM, and the results showed that they can promote wound healing by enhancing angiogenesis and collagen remodeling [102]. A scaffold made of polyethylene terephthalate (PET) can be used to load mesenchymal stromal cells (MSCs) and promote the rate of wound re-epithelialization [103]. BM-MSCs delivered by EGF microspheres into an engineered skin model improved skin wound healing and repaired sweat glands [104]. Stem cells and nano-specific simvastatin, which both have the effect of enhancing wound healing, can be locally applied in the same tissue scaffold (TS) to provide a more effective choice for diabetic wound healing [105]. A DADM/MSC scaffold containing bone marrow mesenchymal stem cells (BM MSCs) and a degenerated decellularized dermal matrix (DADM) promoted wound healing in deep, extensive burn wounds. It was reported that, combined with cell therapy, a pre-synthesized novel nano scaffold made of nanocellulose, type I collagen, and carboxymethyl diethylaminoethyl cellulose has a synergistic effect on wound healing in rats [106]. The researchers developed an in situ cell electrospinning system which overcomes the shortcomings of some stem cell delivery methods, such as a lack of targeting and easy cell loss. The system increased collagen deposition to enhance extracellular matrix remodeling without negatively impacting surface marker expression and the differentiation ability of MSCs.

Furthermore, it also increased the expression of vascular endothelial growth factor (VEGF) and the formation of small blood vessels to promote angiogenesis but significantly reduced the expression of interleukin-6 (IL-6), and ultimately promoted skin wound healing [107]. When comparing poly (ε-caprolactone) (PCL) and poly (ε-caprolactone)/type I collagen (PCol) in the ability to promote biological signaling, wound coverage, and tissue repair processes, it was found that PCol/human Wharton’s jelly mesenchymal stromal cells hWJ-MSCs had a better effect on skin tissue repair [108]. The rheological properties of hydrogels are similar to those of the natural extracellular matrix of skin and they can simulate various functions. Therefore, hydrogels have good development prospects as stem cell delivery vehicles [109]. Hydrogels derived from porcine myocardial matrix have entered clinical trials (NCT02305602) for the prevention and treatment of heart failure after myocardial infarction.

Additionally, natural polymer hydrogels have great advantages in accelerating chronic wound healing. Adipose-derived stem cell (ADSC)- and platelet-rich plasma (PRP)-supported hydrogel systems based on methacrylate gelatin (GelMA) and methacrylate fibroin protein (SFMA) have been developed as cell and growth factor delivery carriers for the treatment of pressure ulcers, and have shown good efficacy [110]. In addition, studies have indicated a great potential for using Poloxam hydrogel as a cell carrier to support human mesenchymal stromal cells (hMSCs) [111]. hUCMSCs, human umbilical cord mesenchymal stem cells encapsulated in a functional injectable thermosensitive hydrogel (chitosan/sodium glycerophosphite/cellulose nanocrystalline, CS/GP/CNC), can be used to repair full-layer skin wounds and significantly accelerate wound closure, microcirculation, tissue remodeling, re-epithelialization, and hair follicle regeneration [112]. Through promoting the transformation of M1-type macrophages into M2-type macrophages and accelerating wound angiogenesis, carboxyethyl chitosan (CEC)-dialdehyde carboxymethyl cellulose hydrogel (MSC-Exos@CEC-DCMC HG) loaded on bone marrow mesenchymal stem cell-derived exosomes (MSC-Exo) resulted in a reduction in inflammation [113]. Microgels consisting of aligned silk nanofibers were used to load MSCs and regulate paracrine signaling; dispersing the MSCs into these injectable silk nanofiber hydrogels can protect and stabilize these cells in wounds.

At the same time, the system is adjustable which enhanced the effect of the MSCs [114]. Recently, a novel polysaccharide-based hydrogel scaffold was made using alginate to create a suitable microenvironment for the delivery of adipose-derived mesenchymal stem cells (ASCs) and was demonstrated to improve wound healing processes and accelerate wound closure [115]. In situ hydrogel systems composed of hyperbranched polyethylene glycol diacrylate (HB-PEGDA) polymers, sulfhydryl functionalized hyaluronic acid (HA-SH), and short RGD peptides bound to adipose-derived stem cells (ASCs) significantly enhanced neovascularization and accelerated wound healing [116]. With the aim of reducing scar formation, bone marrow mesenchymal stem cell (BMSC)-derived nanovesicles (NVs) released by hydrogels were enhanced by genipin and BSA, resulting in efficient ROS clearance and good immunomodulatory activity, and the promoted the proliferation and migration of fibroblasts and vascular endothelial cells, which effectively treated diabetic wounds [117]. Injectable hydrogels from adipose acellular matrix hydrogels (hDAT-gel) combined with human adipose stem cells (hASCs) can accelerate the formation of blood vessels at the wound site to a certain extent and accelerate wound healing, which has great potential in the field of wound healing [118].

It has been found that the stimulation of rat adipose stem cells (rASC) with a 5 μA electrical stimulation (ES), namely 5 μA PFS, can enhance their paracrine function, and delivering the 5 μA PFS with a heparinized PGA–host–guest hydrogel (PGA–Hp hydrogel) effectively accelerated the repair process in a rat full-layer wound model. Amino-functionalized mesoporous silica nanoparticles (MSNs) can enhance the stability of hydrogel beads and significantly improve the proliferative properties of human adipose-derived mesenchymal stem cells (hASCs). In one study, treating circulating monocytes with mesenchymal stem cell (MSC) superserum to produce activated macrophages in a double-layer scaffold composed of hydrogels and nanofibers resulted in faster wound healing rates [119]. In another similar study, a double-layer scaffold composed of hydrogels and nanofibers was also prepared, and ADSCs were inoculated onto it; the optimal performance of re-epithelialization, collagen tissue production, neovascularization, and reduction in inflammation in the wound area were observed [120].

Currently, synthetic cell vectors can be produced by polymerizing acrylic plasma onto medical-grade silica gel for the purpose of delivering hBM-MSCs into the skin [121]. Silk fibroin (SF) was shown to significantly improve the adhesion of bone mesenchymal stem cells (BMSCs), while Col/TSF hybrid scaffolds have excellent skin affinity, good air and water permeability, and good wound healing potential [122].

### 5.2. The Effect of Stem Cell Sheets

Scaffolds have disadvantages such as high requirements for a sufficient supply of cells and correct cell injection positions. In contrast, cell sheets, cells that are either self-supporting or that are delivered from a supporting material but where the material plays no long-term role in the therapy [123], do not need a scaffold and have a higher cell density, which is one of the crucial factors for enhancing the therapeutic function of cell transplantation. Due to their high cell density, cell sheets show a longer retention time at the transplant site as well as more local delivery of growth factors and cytokines.

Cell sheet engineering (CSE) has attracted increasing attention as a competitive alternative to traditional cell-based or stent-based approaches due to its inherent advantages of higher cell survival and biocompatibility [124]. In addition to heart tissue, the benefits of ASC tablets on tissue regeneration are mainly reflected in the skin [125]. Early studies have found that the combination of ASC tablets and artificial skin grafts accelerates wound healing and blood vessel formation [126]. A ROS-induced cell sheet stacking method was designed, and the newly prepared hematoporphyrin was incorporated into a polyketone membrane (Hp-PK membrane), which could improve the delivery efficiency of cell sheets and be effectively applied for wound healing [127]. Human umbilical cord mesenchymal stem cells (hUC-MSCs) were cultivated on Col-T scaffolds to prepare stem cell tablets, which could restore the structure and function of damaged tissues [128]. The injection of disintegrated human amniotic fluid stem cells hAFSC tablets can exert anti-fibrotic properties without delaying wound closure, thereby accelerating skin wound healing and reducing fibrotic scarring, similar to fetal wound healing.

Compared to dissociated cells, treating wound tissues with ASC tablets has the advantages of faster wound healing and minimal risk of long-term side effects [129]. Compared with normal people, diabetic patients with foot ulcers usually show prolonged wound healing due to diabetic neuropathy and blood flow disorders. However, it had been indicated that the direct injection of human fat stem cells (hASCs) can effectively accelerate wound healing in diabetic patients, and this study also pointed out that hASCs have the disadvantage of relative instability [130]. Studies of skin pressure sore healing induced by the injection of MSC-based cell sheets (CSs) in C57Bl/6 mice found that, despite a brief retention of the CSs on the ulcer surface (3–7 days), there was an increase in granulation tissue (GT) thickness and increased vascular maturity, while at the same time, compared to the mesenchymal stromal cell (MSC) exosome group, the CSs had a unique function of skin repair using skin appendages [131]. Additionally, the use of the peritoneum as the support for the precise transplantation of ASC tablets to the back of SD rats had better effects on gross and histopathological repair than that of simply injecting the ASC tablets [132]. Compared with fibroblast tablets, complete tablets composed of amniotic mesenchymal stem cells have a higher tendency to disintegrate, and have the potential to treat burn wounds [133].

Moreover, adipose-derived stem cell (ADSC) tablets can also promote peripheral nerve regeneration [134] and repair ulcerative oral mucosa [135]. In order to harvest cells from a culture medium with intact extracellular matrix (ECM) and preserved intercellular connections, many new materials and methods have been developed, among which nanomaterials are a hot topic. For example, TiO2 nanodots are the most commonly used nanomaterials in photoinduced cell sheet technology, and two-dimensional (2D) nanomaterials such as graphene have also been applied in photoinduced cell sheet technology [136].

### 5.3. Effects of Conditioned Media

There are some limitations to the clinical use of ASCs. For autologous ASCs, ASCs need to be cultured for several weeks to obtain a sufficient number of cells, and the process comes with significant costs and requires staff to maintain the cell culture and cell processing facilities.

At present, there is increasing evidence that mesenchymal stem cells can promote skin repair and regeneration through paracrine actions [137,138,139]. Mesenchymal stem cell-conditioned medium, as a cell-free therapy, can accelerate the wound healing process and avoid the risks of live-cell therapy. Not only that, CM can be easily manufactured, stored, and transported. A study found that Wharton’s jelly mesenchymal stem cell (WJ-MSC)-derived conditioned medium (MSC-CM) secreted some factors that promoted HUVEC multiplication, increased the regeneration of sebaceous glands, and enhanced the angiogenesis induced by human umbilical vein endothelial cells. By promoting the expression of α-SMA, MSC-CM significantly increased the number of skin blood vessels in healed wounds. At the same time, MSCCM was used to treat radiation dermatitis in rats for the first time in [140]. Another similar study showed that WJ-MSCs had a greater ability in sweat gland repair and skin regeneration after skin injury, and the MSC-CM group had the smallest wound area and the highest Col1A2 expression [141]. The conditioned medium of human cord blood mesenchymal stem cells (USC-CM) has an anti-inflammatory effect through the growth factors and cytokines in it, such as EGF [142]. ASC-CM treatment was shown to promote the anti-inflammatory phenotype of macrophages, partially protecting the skin barrier damaged by PMA exposure, and has broad application prospects in wound healing and skin inflammation [143].

Moreover, USC-CM also includes growth factors associated with skin rejuvenation, such as growth differentiation factor-11 (GDF-11) [144]. It has also been found that after injecting a medium conditioned by mesenchymal stromal cells into a wound, a lower inflammation level or enhanced epithelialization were observed [145]. By preparing a diabetic foot ulcer (DFU) model and pretreating the ulcer with rat bone marrow MSC-conditioned medium, it was finally demonstrated that MSC-CM injected into DFU rats promotes the wound healing process by accelerating wound closure, cell proliferation, and angiogenesis, without increasing ulcer cell death [146].

Excessive wound repair can lead to hyperplastic scars or keloids, and it is generally believed that skin fibroblasts play an important role in the scarring process. Hypoxic-conditioned media of placenta-derived mesenchymal stem cells (PMSCs) reduced scarring in vivo and inhibited the proliferation and migration of skin fibroblasts in vitro [147], suggesting that PMSCS may be a promising wound treatment therapy.

### 5.4. Other Practicable Methods to Deliver Stem Cells

Before the transplantation of human adipose-derived stromal cell (hASC) globules (PBM-globules), photobiological regulation can stimulate angiogenesis and tissue regeneration in mouse flaps to improve skin tissue functional recovery. When the wound tissue is treated with cell sheets composed of adipose-derived stem cells (ASCs), more transplanted ASCs can be observed in the wound tissue, and the new skin formed from it has a thickness similar to normal skin and a highly organized collagen structure, which can ultimately improve skin wound healing and reduce scarring [129]. The injection of cell sheets made from disintegrated human amniotic fluid stem cells (hAFSCs) can exert anti-fibrotic properties without delaying wound closure [148].

Studies have found that three-dimensional graphene foam (GF) scaffolds have good biocompatibility, and the combination with bone marrow mesenchymal stem cells (MSCs) promotes the growth and proliferation of MSCs, which can improve skin wound healing [149]. Platelet-rich plasma (PRP) products are believed to have a pro-angiogenic effect and are currently recommended for the treatment of chronic wounds. In one study, the researchers added PRP to irradiated HDMEC and hASC cultures to prevent a large radiation-induced drop in cell numbers, rescuing the proliferation defects caused by external radiation. This method may be beneficial for treating chronic wounds with defective healing processes [150]. Modified acellular dermal matrix (DADM), as a skin substitute combined with bone marrow mesenchymal stem cells (BM-MSCs) implanted on mouse skin wounds, has good survival characteristics and represents a promising alternative therapy for deep, extensive burn wound healing [151]. To verify that preconditioning ASCs with hypoxia leads to enhanced functions of the adipose-derived stem cells (ASCs), porcine adipose stem cells (pASCs) cultured under hypoxia (PASCS-HYP) were transplanted into a mouse model of full-layer skin wound resection and were shown to promote the expression of the angiogenesis marker VegfA and decreased the level of proliferation-promoting Tgfβ1 [152]. Low-level laser therapy (LLLT) can increase the survival rate of ASCs, thereby stimulating the secretion of growth factors [153]. Gene-activated scaffolds (GASs) transfected with genes were loaded with bone marrow-derived mesenchymal stem cells (BM-MSCs) to form GAS/BM-MSCs constructs; these constructs were shown to accelerate the wound healing process and induce in situ regeneration of full-layer skin with sweat glands [154]. The use of micronized amniotic membrane (mAM) as a microcarrier can improve the in vitro expansion efficiency of mesenchymal stem cells [155]. Studies have shown that the pre-treatment of MSCs with bioactive compounds can improve their survival rate and regenerative potential. Human umbilical cord mesenchymal stem cells (hUC-MSCs) can significantly improve healing after treatment with quercetin [156]. Rosuvastatin calcium-loaded scaffolds were prepared and combined with MSCs and implanted into mouse wounds, and it was found that the mesenchymal stem cell–drug scaffolds showed complete skin healing after 30 days [157].

In one study, a 3D radial and vertically aligned nanofiber scaffold was developed that perfectly matched the size, depth, and shape of diabetic wounds to transplant bone marrow mesenchymal stem cells (BMSCs) to promote granulation tissue formation, angiogenesis, and collagen deposition, and tilt the immune response in the pro-regeneration direction [158]. The cord platelet gel (CBPG) developed by the Cord Blood Unit (CBU) has been successfully applied to induce wound closure and tissue regeneration [159].

## 6. Summary and Outlook

In recent years, due to the development of tissue engineering technology, the use of mesenchymal stem cells as a new method for wound healing has garnered extensive research, for uses such as nanotherapy, stem cell therapy, and 3D bioprinting, and due to the low amount of ethical controversy associated with it, there are many tissue sources for mesenchymal stem cells. One of the benefits of mesenchymal stem cells is that bone marrow, adipose tissue, umbilical cord blood, Wharton’s jelly stem cells and amniotic fluid can be easily collected.

In general, the mechanisms through which mesenchymal stem cells promote wound healing have been fully discussed at the molecular level and cellular level. Preclinical and clinical studies on the application of mesenchymal stem cells in the treatment of burns and chronic wounds have fully demonstrated their potential in the field of wound healing. Moreover, mesenchymal stem cells, combined with other tissue engineering techniques, can play a greater role in skin tissue repair. For example, through 3D-printing technology, medical-grade materials such as polycaprolactone (mPCL) and polyethylene terephthalate (PET) can be used to effectively create scaffolds, transporting stem cells to the desired repair site. The scaffold made by combining chitosan and collagen can overcome the limitations of traditional collagen scaffolds and even achieve complete wound healing. In addition, hydrogels can be used as the main raw material for delivery carriers because their rheological properties are similar to those of skin. When combined with emerging materials, mesenchymal stem cells have also shown good performance in promoting wound healing in many studies.

However, many relevant studies are still in the pre-clinical stage, and further exploration is still needed before they are truly transformed into clinical wound healing treatments. However, mesenchymal stem cells have demonstrated their ability, and we have sufficient confidence that mesenchymal stem cells will become the backbone of wound healing in the future.

## Figures and Tables

**Figure 1 biomedicines-12-00743-f001:**
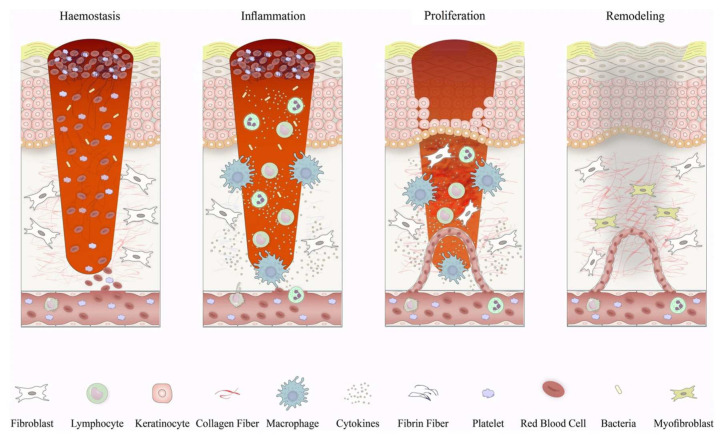
Schematic depiction of wound healing phases and the corresponding cellular responses. In the initial phase of wound healing, when blood clotting occurs, platelets release signaling molecules and chemical messengers that attract inflammatory cells. Inflammation begins with the influx of neutrophils, facilitated by the release of histamine from mast cells. Subsequently, monocytes arrive and differentiate into tissue macrophages, which are responsible for clearing residual cell debris and neutrophils. In the proliferative phase, keratinocytes migrate to bridge the wound, new blood vessels form through the growth of tiny vessels, and specialized cells called fibroblasts replace the initial blood clot with a tissue known as granulation tissue. Macrophages and regulatory T cells play crucial roles during this stage of the healing process. Eventually, the newly formed tissue undergoes further restructuring as fibroblasts reshape the deposited matrix, the blood vessels diminish in size, and specialized cells called myofibroblasts contribute to the overall contraction of the wound. Reproduced with permission from [23].

**Figure 2 biomedicines-12-00743-f002:**
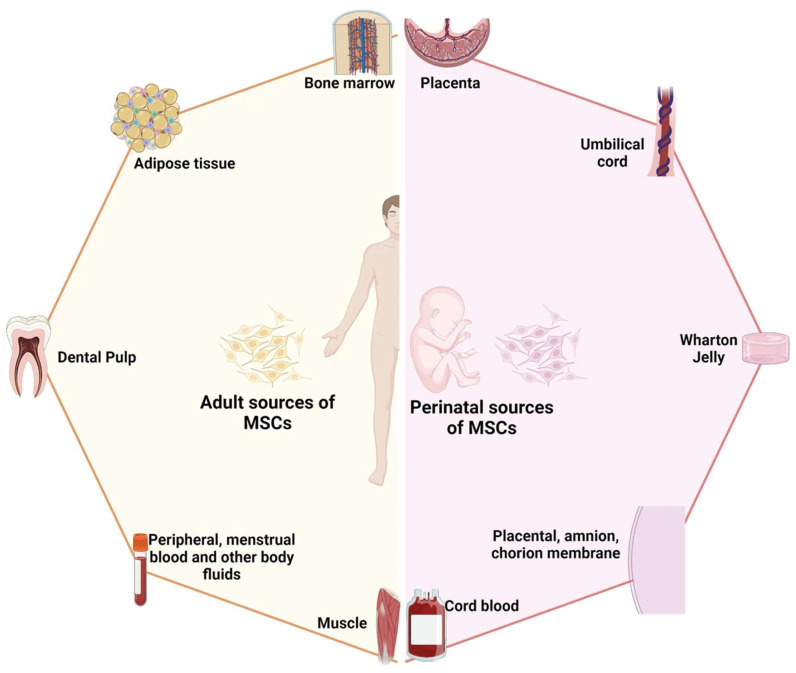
Schematic diagram illustrating the two primary sources of mesenchymal stem cells: adult-derived and perinatal-derived sources. Reproduced with permission from [27].

**Figure 3 biomedicines-12-00743-f003:**
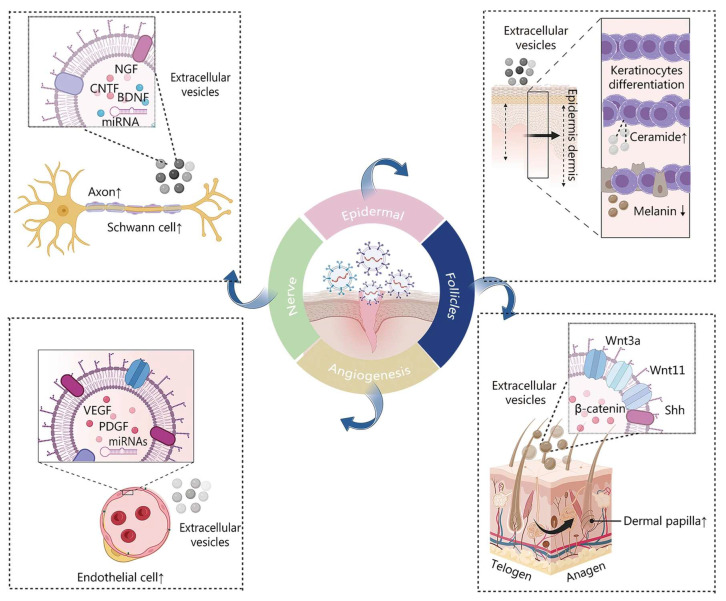
Illustration depicting the impact of MSC-derived EVs on the process of wound healing. Reproduced with permission from [55].

**Table 1 biomedicines-12-00743-t001:** MSC therapy in animal models to promote wound healing.

Condition	Model	Source	Results	Ref.
Severe burn	Rat	Human umbilical cord blood	A reduction in the infiltration of inflammatory cells and the levels of the inflammatory factors IL-1, IL-6, and TNF-α at the wound site, along with increased levels of VEGF and IL-10, contributed to the acceleration of wound healing.	[58]
Excisional wound	Mouse	Human Wharton’s jelly-derived MSCs	Promoted the proliferation and migration of fibroblasts.	[59]
Diabetic foot ulcers	Rat	Bone marrow	Suppressed chronic inflammation, promoted granulation tissue formation and collagen deposition, and stimulated neovascularization.	[60]
Excisional wound	Mouse	Human umbilical cord blood	Differentiated into keratinocytes to promote wound healing	[61]
Full-thickness skin wound	Mouse	Bone marrow	Limited Mφ activation and inflammatory response to improve wound healing by generating TSG-6.	[62]
Thermal burn wound	Mouse	Bone marrow	Promoted wound epithelialization.	[63]
Diabetic wound	Mouse	Human umbilical cord blood	Stimulated collagen deposition and remodeling, and augmented angiogenesis and vessel maturation.	[64]
Full-thickness skin wound	Rat	Adipose tissue	Promoted fibroblast proliferation and migration, as well as suppressed inflammatory response, further enhancing the healing effect.	[65]
Excisional wound	Sheep	Peripheral blood of sheep	No significant effect on the appearance of granulation tissue, neovascularization, or skin adnexa.	[66]

## Data Availability

Not applicable.

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
