# Peer review of "The Role and Prospects of Mesenchymal Stem Cells in Skin Repair and Regeneration"

_biomedicines, 2024, doi:10.3390/biomedicines12040743_

Round 1
Reviewer 1 Report
Comments and Suggestions for Authors
Dear Authors,
Your review idea is good, but the execution is less compelling.
The review, as it is written at the moment, is just a pile of literature citations with no particular purpose or discussion. A good review should be written with some sort of purpose. The purpose can be to draw the attention of researchers to something controversial or encourage researchers in the field to explore something that is still unclear further. Some reviews are written to address an issue or shortcoming in the field or bring up a counterpoint. The review would be greatly improved if you could discuss the benefits vs. shortcomings of various clinical applications of MSC or provide some ideas on future explorations that would benefit the field. For a review to be more reader-friendly, each paragraph and each section should have some introduction and summary. Yours does not have it.
I also want to draw your attention to the proper use of citations. When you present a phenomenon or a mechanism, you can not refer to another review as a source. You have to find the original research manuscript that presented the findings. There are also many instances where you provide a list of mechanisms related to mesenchymal stem cells without citations.
The organization of the review also can be improved. Try to stay focused on one idea. Start by introducing the terminology, continue with the literature-based discussion, and finish with some conclusion or future direction.
I am unsure if it is due to the draft version, but the figure quality was poor on the version I received.
My specific comments can be found in the PDF attached.

I tried to address some grammatical errors, but more thorough editing should be done on the MS to be acceptable for publication.
Author Response
We thank the reviewer for your careful examination of our manuscript, as well as the sincere appreciation. We have improved the quality of our report per the reviewer’s suggestions as detailed below.
---------
Answers to reviewer 1:
We thank the reviewer for your careful examination of our manuscript, as well as the sincere appreciation. We have improved the quality of our report per the reviewer’s suggestions as detailed below.
Reviewers' Comments:
Dear Authors,
Your review idea is good, but the execution is less compelling.
The review, as it is written at the moment, is just a pile of literature citations with no particular purpose or discussion. A good review should be written with some sort of purpose. The purpose can be to draw the attention of researchers to something controversial or encourage researchers in the field to explore something that is still unclear further. Some reviews are written to address an issue or shortcoming in the field or bring up a counterpoint.
1) Please provide for ectodermal origin of MSC
Response: Thank you very much for your careful review and we apologize that this was an oversight in our work. In the latest manuscript, we have reorganized the abbreviations and revised them. Specific changes have been highlighted in the revised manuscript as follows:
MSCs are a multipotent type of stem cells that originate from the mesoderm and ectoderm [1] (1. Introdcution)
- Sheng, G. The developmental basis of mesenchymal stem/stromal cells (MSCs). BMC Dev Biol 2015, 15, 44, doi:10.1186/s12861-015-0094-5.
2) change to accessibility
Response: Thank you for reading our article so carefully and making so many valuable comments. We feel sorry for our carelessness.
MSCs from different sources differ in their accessibility, content, proliferation, im-munomodulatory capacity and cytokine secretion, and have different therapeutic po-tentials for different diseases. (1. Introdcution)
3) Throughout the MS please porvide original reserach citations instead of citing other reviews.
Response: Thank you for reading our article so carefully and making so many valuable comments. We feel sorry for our carelessness, and we have double-checked our manuscript to correct these mistakes. We replace the incorrect review with the appropriate original research manuscript. It is highlighted in the revised manuscript and can be easily found as follows:
Subsequently, many researchers have isolated MSCs from different tissues and have demonstrated that they can differentiate into a wide range of cell types such as osteo-blasts, adipocytes, chondrocytes, tenocytes, cardiomyocytes, keratinocytes, hepatocytes, and neural cells [6,7]. (1. Introdcution)
- Yang, Y.H.; Lee, A.J.; Barabino, G.A. Coculture-driven mesenchymal stem cell-differentiated articular chondrocyte-like cells support neocartilage development. Stem Cells Transl Med 2012, 1, 843-854, doi:10.5966/sctm.2012-0083.
- Yang, Y.K.; Ogando, C.R.; Wang See, C.; Chang, T.Y.; Barabino, G.A. Changes in phenotype and differentiation potential of human mesenchymal stem cells aging in vitro. Stem Cell Res Ther 2018, 9, 131, doi:10.1186/s13287-018-0876-3.
4) not an appropriate citation for the statement
Response: Thank you for your kind reminder. As suggested, we added the missing citations on the statement. This has made our article more informative and we thank you again for your valuable suggestions.
In addition, MSCs have been found to have various biological functions such as im-munomodulation, anti-inflammation, anti-apoptosis, and pro-angiogenesis [8,9]. (1. Introdcution)
- Naji, A.; Eitoku, M.; Favier, B.; Deschaseaux, F.; Rouas-Freiss, N.; Suganuma, N. Biological functions of mesenchymal stem cells and clinical implications. Cell Mol Life Sci 2019, 76, 3323-3348, doi:10.1007/s00018-019-03125-1.
- Tang, H.; Luo, H.; Zhang, Z.; Yang, D. Mesenchymal Stem Cell-Derived Apoptotic Bodies: Biological Functions and Therapeutic Potential. Cells 2022, 11, doi:10.3390/cells11233879.
5) Immune response to collection of MSC form the bone marrow is not mentioned in this reference. Please either remove the statement on immune response or provide an appropriate reference for the statement.
Currently, the most used stem cells are bone marrow-derived mesenchymal stem cells, but they have some limitations, such as a decline in the number and activity of stem cells with age [15]. (1. Introdcution)
- Fu, X.; Liu, G.; Halim, A.; Ju, Y.; Luo, Q.; Song, A.G. Mesenchymal Stem Cell Migration and Tissue Repair. Cells 2019, 8, doi:10.3390/cells8080784.
Response: Thank you for your kind reminder. As suggested, we remove the statement.
6) Provide proper citations for these statements.
Response: Thank you for your kind reminder. As suggested, we added the missing citations on the statement. This has made our article more informative and we thank you again for your valuable suggestions.
Umbilical cord-derived MSCs have been used to treat a wide range of diseases, such as cardiovascular diseases, liver diseases, bone and muscle degenerative diseases, neu-rological injuries, and autoimmune diseases [17-19]. (1. Introdcution)
- Wei, P.; Jia, M.; Kong, X.; Lyu, W.; Feng, H.; Sun, X.; Li, J.; Yang, J.J. Human umbilical cord-derived mesenchymal stem cells ameliorate perioperative neurocognitive disorder by inhibiting inflammatory responses and activating BDNF/TrkB/CREB signaling pathway in aged mice. Stem Cell Res Ther 2023, 14, 263, doi:10.1186/s13287-023-03499-x.
- Wang, C.; Zhao, B.; Zhai, J.; Wang, A.; Cao, N.; Liao, T.; Su, R.; He, L.; Li, Y.; Pei, X.; et al. Clinical-grade human umbilical cord-derived mesenchymal stem cells improved skeletal muscle dysfunction in age-associated sarcopenia mice. Cell Death Dis 2023, 14, 321, doi:10.1038/s41419-023-05843-8.
- Cao, Z.; Wang, D.; Jing, L.; Wen, X.; Xia, N.; Ma, W.; Zhang, X.; Jin, Z.; Shen, W.; Yao, G.; et al. Allogenic Umbilical Cord-Derived Mesenchymal Stromal Cells Sustain Long-Term Therapeutic Efficacy Compared With Low-Dose Interleukin-2 in Systemic Lupus Erythematosus. Stem Cells Transl Med 2023, 12, 431-443, doi:10.1093/stcltm/szad032.
7) Again big statement without citation support.
Response: Thank you for your kind reminder. As suggested, we added the missing citations on the statement. This has made our article more informative and we thank you again for your valuable suggestions.
However, the clinical application of MSCs also faces some challenges and problems, such as cell quality control, immunological rejection, carcinogenic risk, standardized management, etc., which require further research and optimization [20]. (1. Introdcution)
- Zhou, T.; Yuan, Z.; Weng, J.; Pei, D.; Du, X.; He, C.; Lai, P. Challenges and advances in clinical applications of mesenchymal stromal cells. J Hematol Oncol 2021, 14, 24, doi:10.1186/s13045-021-01037-x.
8) This reference is not properly formated. THe title of the manuscript is not provided.
Response: Thank you for your kind reminder. We added the title and it is highlighted in the revised manuscript and can be easily found as follows
- Jeschke, M.G.; Wood, F.M.; Middelkoop, E.; Bayat, A.; Teot, L.; Ogawa, R.; Gauglitz, G.G. Scars. Nat Rev Dis Primers 2023, 9, 64, doi:10.1038/s41572-023-00474-x. (References)
9) The figure legend is very limited and not explaining what is occuring in each phase. For example, what is happening on the haemostasis pahse? WHat is proliferating in the proliferation phase? What is happening in the inflammation phase? and so on. Wihtout explanation this figure has no meaning.
Response: Thank you for your kind reminder. As suggested, we added the detailed explanation of changes in different stages as follows:
Figure 1. Schematic depiction of wound healing phases and the corresponding cellular responses. In the initial phase of wound healing, when blood clotting occurs, platelets release signaling molecules and chemical messengers that promote the attraction of inflammatory cells. Inflammation begins with the influx of neutrophils, facilitated by the release of histamine from mast cells. Subsequently, monocytes arrive and differentiate into tissue macrophages, responsible for clearing residual cell debris and neutrophils. In the proliferative phase, keratinocytes migrate to bridge the wound, new blood vessels form through the growth of tiny vessels, and specialized cells called fibroblasts replace the initial blood clot with a tissue known as granulation tissue. Macrophages and regulatory T cells play crucial roles during this stage of the healing process. Eventually, the newly formed tissue undergoes further restructuring as fibroblasts reshape the deposited matrix, blood vessels diminish in size, and specialized cells called myofibroblasts contribute to the overall con-traction of the wound. Reproduced with permission [21]. (Figure 1)
10) Who are they? New-born blood vessels?
Response: Thank you for your kind reminder. As suggested, we have corrected as follows:
The new-born blood vessels also release some cytokines, such as vascular endothelial growth factor (VEGF), transforming growth factor-β (TGF-β), IL-6, etc., which regulate the inflammatory response and fibrotic process, inhibit the activation and secretion of inflammatory cells, and promote the proliferation of fibroblasts and collagen synthesis [29,32]. (2. The Wound Healing-promoting Mechanisms of Mesenchymal Stem Cells)
The new-born blood vessels can influence the remodeling process of wound tissue, promoting steps such as epithelial formation, collagen remodeling, fibrosis and the formation of structures such as granulation tissue and scar tissue [34,35]. (2. The Wound Healing-promoting Mechanisms of Mesenchymal Stem Cells)
11) Repetition of the same content form 131-133
Response: Thank you for your kind reminder. As suggested, we have corrected as follows:
In recent years, it has been demonstrated that human umbilical cord mesenchymal stem cell exosomes could also accelerate re-epithelialization of burn wounds by in-creasing the phosphorylation level of AKT and enhancing the downstream effects of AKT signaling, thereby promoting the proliferation and migration of keratin-forming cells [44]. The AKT signaling pathway is a signal transduction pathway involved in cell survival and proliferation, which plays an important role in skin regeneration [46]. (2. The Wound Healing-promoting Mechanisms of Mesenchymal Stem Cells)
12) This sentence has incorrect grammar. Woulds do not proliferate. Please correct.
Response: Thank you for your kind reminder. As suggested, we have corrected as follows:
Fibroblasts are the main cell type in the dermis of the skin and they play an important role in the proliferation and remodeling stages of wounds [47]. (2. The Wound Healing-promoting Mechanisms of Mesenchymal Stem Cells)
13) Reduce scar formation?
Response: Thank you for your kind reminder. As suggested, we have corrected as follows:
Studies have shown that human adipose MSC exosomes can reduce scar formation in burn wounds by decreasing the expression and activity of TGF-β and inhibiting the phosphorylation and nuclear translocation of Smad2/3, thereby reducing the proliferation and collagen synthesis of scar fibroblasts [51]. (2. The Wound Healing-promoting Mechanisms of Mesenchymal Stem Cells)
14) This paragraph needs to be split and distributed in the paragraph above such that the line of evidence supporting re-epithelialization is continuous and the line on neovascularization is continuous. The definition of exosome is mentioned here again which is redundant.
Response: Thank you for your kind reminder. As suggested, we have deleted the statement.
15) This whole sentience needs to have citations to every disadvantage listed.
Response: Thank you for your kind reminder. As suggested, we added the missing citations on the statement.
However, all of these methods have certain limitations and side effects, such as deb-ridement that can cause infection and bleeding [68], etc. Therefore, there is an urgent need to explore some new ways. (3. The Potential of Mesenchymal Stem Cells in the Treatment of Clinical Diseases)
- Tang, J.; Guan, H.; Dong, W.; Liu, Y.; Dong, J.; Huang, L.; Zhou, J.; Lu, S. Application of Compound Polymyxin B Ointment in the Treatment of Chronic Refractory Wounds. Int J Low Extrem Wounds 2022, 21, 320-324, doi:10.1177/1534734620944512.
16) I don't understand why this sentence is here. It adds nothing to the MSC.
Response: Thank you for your kind reminder. As suggested, we have deleted the statement.
17) Induced
Response: Thank you for your kind reminder. As suggested, we added the missing word.
Except chronological aging, human skin also undergoes photoaging, a kind of sun-induced skin aging. (4. Applications of Mesenchymal Stem Cells in Skin Regeneration and Rejuvenation.)
18) Insert our understanding of infront of molecular mechanisms
Response: Thank you for your kind reminder. As suggested, we added the current molecular mechanism.
Actually, in the past 30 years, the molecular mechanism of skin photoaging such as production of intracellular ROS has made substantial progress. (4. Applications of Mesenchymal Stem Cells in Skin Regeneration and Rejuvenation.)
19) incorrect wording here
Response: Thank you for your kind reminder. As suggested, we replaced the wrong word by proven.
The exposure of ultraviolet rays for a long time is proven to decrease the collagen protein production. (4. Applications of Mesenchymal Stem Cells in Skin Regeneration and Rejuvenation.)
20) this topic was already discussed above. It is reduntant and does not belong here. one word
Response: Thank you for your kind reminder. As suggested, we deleted the statement.
21) IT is unclear from his summary if the stem cells were incorporated into the matrix or injected IV separately.
Response: Thank you for your kind reminder. As suggested, we revised the statement as follows:
Transplanting human gingival tissue pluripotent mesenchymal stem cells/stromal cells reduced into 3D printed medical grade polycaprolactone (mPCL) dressings wound contracture and significantly improved skin regeneration through granulation and re-epithelialization [95]. (5.1. Stem Cell Delivery in Skin Wounds)
22) Cell tablets is not a familiar term in medicine and hence it needs to be better explained here. Also have to provide reference for this statement
Response: Thank you for your kind reminder. As suggested, we revised the statement as follows:
In contrast, cell sheets, cells that are either self-supporting or that are delivered from a supporting material but where the material plays no long-term role in the therapy[120], do not need a scaffold but has a higher cell density which is one of the crucial factors to enhance the therapeutic function of cell transplantation. (5.2. The Effect of Stem Cell Sheets)
23) Need to clarify what cell slice engineering means and how it is done. with references.
Response: Thank you for your kind reminder. As suggested, we revised the statement as cell sheet engineering which is the same meaning with the cell sheet:
Cell sheet engineering (CSE) has attracted increasing attention as a competitive alternative to traditional cell-based or stent-based approaches due to its inherent ad-vantages of higher cell survival and biocompatibility [121]. (5.2. The Effect of Stem Cell Sheets)
24) This sentence is confusing. The meaning of unique appendage for skin repair sounds strange and unclear.
Response: Thank you for your kind reminder. As suggested, we revised the statement as follows:
Compared with normal people, diabetic patients with foot ulcers usually show pro-longed wound healing by the reason of diabetic neuropathy and blood flow disorders. However, it had been indicated that direct injection of human fat stem cells (hASC) can effectively accelerate the wound healing of diabetic patients, but this study also points out that hASC has the disadvantage of relative instability [127]. (5.2. The Effect of Stem Cell Sheets)
25) Need to provide reference for this citation..
Response: Thank you for your kind reminder. As suggested, we added the reference as follows:
At present, there is increasing evidence that mesenchymal stem cells promote skin repair and regeneration through paracrine action [134-136]. (5.3. Effects of Conditioned Medias)
- Zhang, J.; Huang, X.; Wang, H.; Liu, X.; Zhang, T.; Wang, Y.; Hu, D. The challenges and promises of allogeneic mesenchymal stem cells for use as a cell-based therapy. Stem Cell Res Ther 2015, 6, 234, doi:10.1186/s13287-015-0240-9.
- Dittmer, J.; Leyh, B. Paracrine effects of stem cells in wound healing and cancer progression (Review). Int J Oncol 2014, 44, 1789-1798, doi:10.3892/ijo.2014.2385.
- Ratajczak, M.Z.; Kucia, M.; Jadczyk, T.; Greco, N.J.; Wojakowski, W.; Tendera, M.; Ratajczak, J. Pivotal role of paracrine effects in stem cell therapies in regenerative medicine: can we translate stem cell-secreted paracrine factors and microvesicles into better therapeutic strategies? Leukemia 2012, 26, 1166-1173, doi:10.1038/leu.2011.389.
26) How is this citation relates to the CM?
Response: Thank you for your kind reminder. It is right that the citation is unrelated to the CM. Therefore, we deleted the statement.

Reviewer 2 Report
Comments and Suggestions for Authors
I have just few minor comments. I suggest to expand the future perspective of stem cells use as well as the delivery approach
Comments on the Quality of English Languagefew english errors
Author Response
We thank the reviewer for your careful examination of our manuscript, as well as the sincere appreciation. We have improved the quality of our report per the reviewer’s suggestions as detailed below.

Reviewer 3 Report
Comments and Suggestions for Authors
In this Review article, Wu and coworkers summarized the current knowledge and the most recent achievements in skin regeneration mediated by mesenchymal stem cells. The topic is interesting for Biomedicines readers, although several review articles are already available on this topic. Nevertheless, the Authors succeeded in writing a brief, comprehensive and useful outline of the state-of-art in this research field. Indeed, the paper is well-designed and written in clear, good English. The quality of figures and tables is suitable for the chosen journal target. The reference list is up-to-date and exhaustive. As a minor suggestion, in this reviewer opinion, it is recommended to shorten section 3 to focus the reader attention on section 5, which is the actual frontier presented in the review, also considering the interests of Biomedicines readers.
Author Response

(The authors gave the same response as above.)
